# Highly Bright Silica-Coated InP/ZnS Quantum Dot-Embedded Silica Nanoparticles as Biocompatible Nanoprobes

**DOI:** 10.3390/ijms231810977

**Published:** 2022-09-19

**Authors:** Kyeong-Min Ham, Minhee Kim, Sungje Bock, Jaehi Kim, Wooyeon Kim, Heung Su Jung, Jaehyun An, Hobeom Song, Jung-Won Kim, Hyung-Mo Kim, Won-Yeop Rho, Sang Hun Lee, Seung-min Park, Dong-Eun Kim, Bong-Hyun Jun

**Affiliations:** 1Department of Bioscience and Biotechnology, Konkuk University, Seoul 05029, Korea; 2Company of Global Zeus, Hwaseong 18363, Korea; 3Company of BioSquare, Hwaseong 18449, Korea; 4AI-Superconvergence KIURI Translational Research Center, Ajou University School of Medicine, Suwon 16499, Korea; 5School of International Engineering and Science, Jeonbuk National University, Jeonju 54896, Korea; 6Department of Chemical and Biological Engineering, Hanbat University, Daejeon 34158, Korea; 7Department of Urology, Stanford University School of Medicine, Stanford, CA 94305, USA

**Keywords:** quantum dots (QDs), silica-coated InP/ZnS QD-embedded silica nanoparticles, biocompatible nanoprobes, photoluminescence (PL), syngeneic mice, in vivo, bioimaging

## Abstract

Quantum dots (QDs) have outstanding optical properties such as strong fluorescence, excellent photostability, broad absorption spectra, and narrow emission bands, which make them useful for bioimaging. However, cadmium (Cd)-based QDs, which have been widely studied, have potential toxicity problems. Cd-free QDs have also been studied, but their weak photoluminescence (PL) intensity makes their practical use in bioimaging challenging. In this study, Cd-free QD nanoprobes for bioimaging were fabricated by densely embedding multiple indium phosphide/zinc sulfide (InP/ZnS) QDs onto silica templates and coating them with a silica shell. The fabricated silica-coated InP/ZnS QD-embedded silica nanoparticles (SiO_2_@InP QDs@SiO_2_ NPs) exhibited hydrophilic properties because of the surface silica shell. The quantum yield (QY), maximum emission peak wavelength, and full-width half-maximum (FWHM) of the final fabricated SiO_2_@InP QDs@SiO_2_ NPs were 6.61%, 527.01 nm, and 44.62 nm, respectively. Moreover, the brightness of the particles could be easily controlled by adjusting the amount of InP/ZnS QDs in the SiO_2_@InP QDs@SiO_2_ NPs. When SiO_2_@InP QDs@SiO_2_ NPs were administered to tumor syngeneic mice, the fluorescence signal was prominently detected in the tumor because of the preferential distribution of the SiO_2_@InP QDs@SiO_2_ NPs, demonstrating their applicability in bioimaging with NPs. Thus, SiO_2_@InP QDs@SiO_2_ NPs have the potential to successfully replace Cd-based QDs as highly bright and biocompatible fluorescent nanoprobes.

## 1. Introduction

Quantum dots (QDs), a type of colloidal semiconductor nanocrystal, have been applied in various bio-fields owing to their good optical properties, such as high fluorescence intensity, low photobleaching, wide absorbance wavelengths, and narrow emission wavelengths, compared to conventional organic fluorescent materials (e.g., organic dyes and fluorescent proteins) [1,2,3,4,5,6]. Among various QDs, cadmium selenide (CdSe)-based QDs have been the most widely studied, owing to their advantages such as high quantum yield (QY), particle stability, and a photoluminescent (PL) emission range across almost the entire visible light region. However, it is well known that cadmium and selenide ions within the CdSe-based QDs could cause serious problems in terms of environmental hazards and toxicity to organisms [7,8,9].

Indium phosphide (InP) QDs, which have been widely studied as Cd-free QDs, were less toxic and harmful to the environment and organisms than CdSe-based QDs [10,11]. In addition, InP QDs have a bulk bandgap of 1.35 eV and an exciton Bohr radius of approximately 10 nm, which enables QDs to have PL emission wavelengths ranging from visible light (blue) to the near infrared [12,13,14]. However, InP QDs typically exhibit a poor QY of <1%, which is attributed to the surface trap states of InP QDs [15,16]. To overcome the low QY of InP QDs, InP-based QDs with an InP core and a shell structure consisting of higher energy bandgap materials such as zinc sulfide (ZnS) were synthesized [17]. The ZnS shell passivated surface defects, prevented the oxidation of the InP core, and significantly increased the QY of InP/ZnS QDs to about 40% [18,19]. Although InP/ZnS QDs possess the potential to be used in biological applications [20,21,22,23,24,25], compared to the well-developed CdSe-based QDs, these still have a lower brightness, which limits the direct application of single InP/ZnS QDs for bioimaging [26,27,28].

To overcome the low brightness of single QDs, approaches for embedding multiple QDs onto the surface of a silica template structure have been leveraged [29,30,31]. As the fabricated silica template-based multi-QDs are brighter than single QDs, they could be used as advanced strategies for applying QDs in biological fields. Among these approaches, silica-coated QD-embedded silica nanoparticles (SiO_2_@QDs@SiO_2_ NPs) have many structural advantages that are suited for biological applications [32,33,34,35,36]. The fabrication process of SiO_2_@QDs@SiO_2_ NPs yields an efficient assembly of approximately 500 QDs on a single silica template. The fabricated SiO_2_@QDs@SiO_2_ NPs exhibited a 200-fold stronger PL emission than those of single QDs. In addition, the silica shell, which is located on the surface of the SiO_2_@QDs@SiO_2_ NPs, ensures a good colloidal stability in hydrophilic solvents and facilitates surface modification. A strong fluorescence signal from the SiO_2_@QDs@SiO_2_ NP-tagged cells was observed, and the suitability of SiO_2_@QDs@SiO_2_ NPs for bioimaging applications was confirmed.

Most studies focus on the fabrication of silica template-based multi-QDs for bioimaging using Cd-based QDs [37,38,39]. However, there has been little progress in the fabrication of silica template-based multi-QDs using InP/ZnS QDs [40,41] and their applications in bioimaging. Miao et al. synthesized Hsp 90α-functionalized mesoporous silica NP-InP/ZnS QD complexes and used them for screening proteins and real-time cell imaging [42]. Perton et al. synthesized polysaccharide-coated stellate mesoporous silica-InP/ZnS QDs and used them for the in vivo fluorescent imaging of zebrafish [43]. However, detailed approaches to materials have not been studied well.

In this study, highly bright and biocompatible bioimaging nanoprobes were fabricated by densely embedding multiple InP/ZnS QDs onto the surfaces of silica templates and coating them with a silica shell. The fabricated silica-coated InP/ZnS QD-embedded SiO_2_ NPs (SiO_2_@InP QDs@SiO_2_ NPs) exhibited hydrophilicity owing to the silica shell on the particle surface. In addition, SiO_2_@InP QDs@SiO_2_ NPs showed a much stronger PL intensity than single hydrophilic CdSe/ZnS QDs, and these strong fluorescence signals could be advantageously applied to bioimaging. Furthermore, the brightness control of the SiO_2_@InP QDs@SiO_2_ NPs was performed by adjusting the amount of added InP/ZnS QDs during the particle fabrication process. To confirm the biological applicability, cytotoxicity investigation, in vivo biodistribution, and fluorescence imaging were performed, thereby proving that SiO_2_@InP QDs@SiO_2_ NPs can be utilized in the field of bioimaging as an alternative to CdSe-based QDs.

## 2. Results and Discussion

### 2.1. Fabrication of SiO_2_@InP QDs@SiO_2_ NPs

The fabrication flow of the proposed, bright, and biocompatible silica-coated InP/ZnS QD-embedded silica nanoparticles (SiO_2_@InP QDs@SiO_2_ NPs) is illustrated in Figure 1a. SiO_2_ NPs, which were used as templates, were synthesized using a sol–gel process based on the Stöber method [44]. The surface of SiO_2_ NPs was modified to a thiol (-SH) group, which has high affinity for QDs, by using 3-mercaptopropyltrimethoxysilane (MPTS). Owing to the affinity between the -SH group and InP/ZnS QDs, several InP/ZnS QDs were embedded onto the surface of the thiol-modified SiO_2_ NPs. The addition of MPTS and NH_4_OH after embedding increased the number of InP/ZnS QDs embedded on the surface of the thiol-modified SiO_2_ NPs [28]. To increase biocompatibility and to prevent the leaching of embedded InP/ZnS QDs on SiO_2_ NPs, these were coated with silica shells by reacting with tetraethyl orthosilicate (TEOS) and NH_4_OH.

Transmission electron microscopy (TEM) images of SiO_2_ NPs, InP/ZnS QDs, and SiO_2_@InP QDs@SiO_2_ NPs were obtained to confirm the morphology and size of each particle (Figure 1b). SiO_2_ NPs showed a uniform spherical shape with a size of 172.2 ± 7.2 nm (Figure 1b(i)). The size of the InP/ZnS QDs (Mesolight, Suzhou, China) was estimated to be 5.1 ± 1.0 nm (Figure 1b(ii)). The SiO_2_@InP QDs@SiO_2_ NPs were fabricated with similar morphologies and had a final size of 201.4 ± 9.9 nm (Figure 1b(iii)). The fabricated SiO_2_@InP QDs@SiO_2_ NPs showed that approximately 1200–1500 InP/ZnS QDs were densely embedded onto the SiO_2_ NPs, and silica shells were formed on the surfaces of the SiO_2_@InP QDs NPs (Figure 1b(iv)).

### 2.2. Characterization of SiO_2_@InP QDs@SiO_2_ NPs

To evaluate the photophysical properties of the SiO_2_@InP QDs@SiO_2_ NPs, the UV/Vis/NIR absorbance spectra of the SiO_2_ NPs, InP/ZnS QDs, and SiO_2_@InP QDs@SiO_2_ NPs were measured (Figure 2a). The absorbance measurement range was 300–1100 nm. When the absorbance was measured, SiO_2_ NPs and SiO_2_@InP QDs@SiO_2_ NPs were measured at the same concentration (0.1 mg/mL), and InP/ZnS QDs were measured at 0.07 mg/mL, which was based on the amount of added QDs during the fabrication of the SiO_2_@InP QDs@SiO_2_ NPs. An absorbance analysis showed that the absorbance of the fabricated SiO_2_@InP QDs@SiO_2_ NPs was higher compared to those of the SiO_2_ NPs and InP/ZnS QDs over the wavelength range of UV/Vis. In addition, both the InP/ZnS QDs and SiO_2_@InP QDs@SiO_2_ NPs showed an absorption peak at approximately 500 nm. The absorption spectrum of the SiO_2_@InP QDs@SiO_2_ NPs showed that the InP/ZnS QDs were well embedded on the surface of the thiol-modified SiO_2_ NPs, and that the SiO_2_@InP QDs@SiO_2_ NPs maintained the absorption property of the InP/ZnS QDs.

To evaluate the luminous efficiency of the SiO_2_@InP QDs@SiO_2_ NPs, the quantum yields (QYs) of the InP/ZnS QDs and SiO_2_@InP QDs@SiO_2_ NPs were compared. The QY of the InP/ZnS QDs was 15.02%, and that of SiO_2_@InP QDs@SiO_2_ NPs was 6.61% (Figure 2b). Processes such as modifying the surface of QDs and coating silica shells can affect the QYs of QDs [45,46]. These results were also observed in previous studies related to silica-template-based multi-QDs [32,33,36].

To evaluate the emission properties of the SiO_2_@InP QDs@SiO_2_ NPs, the PL spectra of InP/ZnS QDs and SiO_2_@InP QDs@SiO_2_ NPs were compared (Figure 2c). The maximum emission peak wavelength of InP/ZnS QDs and SiO_2_@InP QDs@SiO_2_ NPs were 525.09 nm and 527.01 nm, respectively. The emission peaks of the SiO_2_@InP QDs@SiO_2_ NPs shifted little from the emission peak of InP/ZnS QDs. The full-width half-maximum (FWHM) of the InP/ZnS QDs and the SiO_2_@InP QDs@SiO_2_ NPs were 41.38 nm and 44.62 nm, respectively. The FWHM values were not significant changed.

The CdSe/ZnS QDs have been most widely used in biological applications because of their excellent QY and photostability [47,48,49]. Therefore, they were set as a comparison group for the SiO_2_@InP QDs@SiO_2_ NPs. The selected CdSe/ZnS QDs exhibit a photoluminescence (PL) emission wavelength range similar to that of the InP/ZnS QDs (Appendix A). To compare the QY under aqueous conditions, the hydrophobic ligands of the CdSe/ZnS QDs were replaced with hydrophilic ligands [50]. The QY of the hydrophilic CdSe/ZnS QDs was 87.52% (Appendix A). As the SiO_2_@InP QDs@SiO_2_ NPs were fabricated based on InP/ZnS QDs with a low QY, they had a lower QY than the hydrophilic CdSe/ZnS QDs.

To evaluate the luminous intensity, the PL intensities of the SiO_2_@InP QDs@SiO_2_ NPs and the hydrophilic CdSe/ZnS QDs were compared. The PL spectra of the SiO_2_@InP QDs@SiO_2_ NPs and the hydrophilic CdSe/ZnS QDs with the same particle concentration (2.66 × 10^12^ particles/mL) were compared in the visible light region (inset of Appendix A). When each of the particles were irradiated by a light source with an excitation wavelength of 385 nm, the PL intensity of the SiO_2_@InP QDs@SiO_2_ NPs in the 500–550 nm emission wavelength range was much stronger than that of the hydrophilic CdSe/ZnS QDs. At an emission wavelength of 527 nm, the SiO_2_@InP QDs@SiO_2_ NPs exhibited the maximum fluorescence signal, and the maximum PL intensity of the SiO_2_@InP QDs@SiO_2_ NPs was up to 98.4 times higher than that of the hydrophilic CdSe/ZnS QDs (Appendix A). Although the QY of the SiO_2_@InP QDs@SiO_2_ NPs was lower than that of the hydrophilic CdSe/ZnS QDs, the SiO_2_@InP QDs@SiO_2_ NPs were brighter than the hydrophilic CdSe/ZnS QDs at the same particle concentration because the SiO_2_@InP QDs@SiO_2_ NPs contained multiple InP/ZnS QDs. With these optical properties, SiO_2_@InP QDs@SiO_2_ NPs are advantageous for applications in fluorescence bioimaging.

To investigate the hydrophilicity of the SiO_2_@InP QDs@SiO_2_ NPs, the fabricated SiO_2_@InP QDs@SiO_2_ NPs were dispersed in distilled water (DW), and an equal volume of chloroform (CHCl_3_) was added. The mixture was vortexed for a few minutes and photographed under daylight and UV light after the phase was separated (Figure 2d). As a result, the SiO_2_@InP QDs@SiO_2_ NPs with a hydrophilic silica shell on the surface were well dispersed in DW, but not in CHCl_3_. These features indicate that the SiO_2_@InP QDs@SiO_2_ NPs had hydrophilic properties.

### 2.3. Brightness Control of SiO_2_@InP QDs@SiO_2_ NPs

To evaluate the brightness control of the SiO_2_@InP QDs@SiO_2_ NPs, SiO_2_@InP QDs@SiO_2_ NPs with different numbers of embedded QDs were fabricated, and their brightnesses were compared. The SiO_2_@InP QDs@SiO_2_ NPs, which were fabricated by varying the amount of added QDs (0, 0.0875, 0.175, 0.35, and 0.7 mg of QDs per 1 mg of SiO_2_ NPs) were analyzed by using TEM (Figure 3a). As the amount of added QDs increased from 0 mg to 0.7 mg per 1 mg of SiO_2_ NPs, QDs were densely embedded onto the surface of the SiO_2_ NPs. When the QDs exceeded 0.7 mg, excess QDs aggregated and did not embed onto the surface of the SiO_2_ NPs (Appendix A).

To evaluate the luminous variation according to the number of QDs in the SiO_2_@InP QDs@SiO_2_ NPs, the PL spectra of the SiO_2_@InP QDs@SiO_2_ NPs with different numbers of embedded QDs were compared in the visible light region (Figure 3b). As the amount of added QDs increased, the number of embedded QDs on the surface of the SiO_2_ NPs increased. Consequently, the PL intensity of the SiO_2_@InP QDs@SiO_2_ NPs increased. At an emission wavelength of 527 nm, the SiO_2_@InP QDs@SiO_2_ NPs exhibited maximum fluorescence, and the maximum PL intensity of the SiO_2_@InP QDs@SiO_2_ NPs also increased proportionally with the amount of added QDs (Figure 3c). Based on these results, the brightness of the SiO_2_@InP QDs@SiO_2_ NPs could be easily controlled by adjusting the amount of added QDs.

### 2.4. Cytotoxicity Investigation and In Vivo Biodistribution of SiO_2_@InP QDs@SiO_2_ NPs

To evaluate the suitability of the SiO_2_@InP QDs@SiO_2_ NPs for bioimaging applications, we first tested the cytotoxicity of these NPs in human cells (Appendix A). SiO_2_@InP QDs@SiO_2_ NPs and hydrophilic CdSe/ZnS QDs (3.56 × 10^11^–1.39 × 10^9^ and 1.96 × 10^14^–7.66 × 10^11^ particles/mL, respectively) were used to treat human lung cancer (A549) cells, and cell viability was determined after 24 h. When A549 cells were treated with the highest concentration of SiO_2_@InP QDs@SiO_2_ NPs or hydrophilic CdSe/ZnS QDs, cell viability was not significantly affected compared to the untreated cells (87.9 ± 6.2% and 94.2 ± 5.4% in SiO_2_@InP QDs@SiO_2_ NPs and hydrophilic CdSe/ZnS QDs, respectively). Thus, the SiO_2_@InP QDs@SiO_2_ NPs showed no significant cytotoxicity against human cells, which is appropriate for bioimaging applications in vivo.

To assess the applicability of SiO_2_@InP QDs@SiO_2_ NPs in bioimaging tumors in vivo, tumor syngeneic mice were intravenously administered SiO_2_@InP QDs@SiO_2_ NPs and hydrophilic CdSe/ZnS QDs (2.22 × 10^11^ particles/mL each) via the tail vein. At 24 h after intravenous administration, the biodistributions of the SiO_2_@InP QDs@SiO_2_ NPs and hydrophilic CdSe/ZnS QDs in major organs (liver, lung, kidney, and spleen) and in tumors of tumor syngeneic mice were monitored by measuring the fluorescence signal of the QDs using an IVIS imaging system (Figure 4a). Owing to the enhanced permeability and retention (EPR) effect, both SiO_2_@InP QDs@SiO_2_ NPs and hydrophilic CdSe/ZnS QDs prominently accumulated at the tumor site compared to those in other major organs [51,52,53]. To investigate the residual fluorescence signal of particles accumulated in the tumor, the average radiant efficacy in major organs and tumors was measured (Figure 4b). The SiO_2_@InP QDs@SiO_2_ NPs showed significantly higher fluorescence in the tumor tissue compared to those in other organs and in the untreated control. More importantly, the retention of the SiO_2_@InP QDs@SiO_2_ NPs at the tumor site was clearly observed, with a similar or comparable efficiency to that of the hydrophilic CdSe/ZnS QDs. Hence, we suggest that SiO_2_@InP QDs@SiO_2_ NPs can be utilized as an alternative to CdSe-based QDs for bioimaging, owing to their outstanding fluorescence signal and excellent biocompatibility.

## 3. Materials and Methods

### 3.1. Materials

Indium phosphide/zinc sulfide quantum dots (InP/ZnS QDs, λ_em_. 527 nm) were purchased from Mesolight (Suzhou, China). Cadmium selenide/zinc sulfide quantum dots (CdSe/ZnS QDs, λ_em_. 530 nm) were purchased from ZEUS (Hwaseong, Korea). Tetraethyl orthosilicate (TEOS), 3-mercaptopropyltrimethoxysilane (MPTS), and dichloromethane (DCM) were purchased from Samchun (Pyeongtaek, Korea). Ethyl alcohol (EtOH, 99.9%) and aqueous ammonium hydroxide (NH_4_OH, 27%) were purchased from Daejung (Siheung, Korea). Chloroform (CHCl_3_, 99%), tetramethylammonium hydroxide pentahydrate (TMAH, 97%), 3-mercaptopropionic acid (MPA, 99%), paraformaldehyde, and crystal violet were purchased from Sigma-Aldrich (St. Louis, MO, USA). A549 (human non-small cell lung cancer, CCL-185) and 4T1 (mouse breast cancer, CRL-2539) cells were purchased from the American Type Culture Collection (ATCC, Manassas, VA, USA). High-glucose Dulbecco’s modified Eagle medium (DMEM) and fetal bovine serum (FBS) were purchased from Biowest (Nuaille, France). The penicillin–streptomycin solution was purchased from Welgene (Daegu, Korea). Sodium dodecyl sulfate (SDS) was purchased from LPS Solution (Daejeon, Korea). Phosphate-buffered saline (PBS; pH 7.4) was purchased from BYLABS (Hanam, Korea). Eight-week-old male BALB/c mice were purchased from Orient Bio, Inc. (Seongnam, Korea).

### 3.2. Preparation of Thiol-Modified Silica Nanoparticles

Silica nanoparticles (SiO_2_ NPs) with an average diameter of approximately 172 nm were prepared using a sol–gel process based on the Stöber method [44]. TEOS (1.6 mL) was mixed with 40 mL of EtOH. NH_4_OH (3.0 mL) was added to the mixture while stirring, and the mixture was allowed to react at room temperature for 20 h while stirring at 700 rpm. The SiO_2_ NPs were centrifuged for 15 min at 8500 rpm and washed 5 times with EtOH. After 1 mg of SiO_2_ NPs were dispersed in a microtube with 980 μL of EtOH, 10 μL of distilled water, 10 μL of MPTS, and 2.5 μL of NH_4_OH, the mixture was incubated at 50 °C for 1 h. The thiol-group-introduced SiO_2_ NPs were harvested after centrifugation and washed thrice with EtOH to remove the excess reagents.

### 3.3. Fabrication of Silica-Coated InP/ZnS QD-Embedded SiO_2_ NPs (SiO_2_@InP QDs@ SiO_2_ NPs)

Thiol-modified SiO_2_ NPs (1 mg in 100 μL EtOH) and 5 μL distilled water were added to 400 μL DCM and mixed with 0.7 mg of InP/ZnS QDs (25 mg/mL in toluene). The mixture was then incubated at room temperature for 1 h, with sonication for 2 min every 30 min. Next, 5 μL MPTS and 5 μL NH_4_OH were added to the mixture, and the mixture was incubated at room temperature for 1 h. InP/ZnS QD-embedded SiO_2_ NPs (SiO2@InP QDs NPs) were centrifuged for 10 min at 8500 rpm and washed thrice with EtOH. The washed SiO_2_@InP QDs NPs were dispersed in 500 μL EtOH. After dispersion, 5 μL TEOS and 5 μL NH_4_OH were added to the solution. The mixture was then incubated at room temperature for 20 h. Silica-coated InP/ZnS QD-embedded SiO_2_ NPs (SiO_2_@InP QDs@SiO_2_ NPs) were centrifuged for 10 min at 8500 rpm and washed thrice with EtOH. The SiO_2_@InP QDs@SiO_2_ NPs were then dispersed in EtOH to adjust the concentration to 1 mg/mL.

### 3.4. Surface Modification of CdSe/ZnS QDs

Hydrophilic CdSe/ZnS QDs were prepared by replacing hydrophobic surface ligands (oleic acid) with hydrophilic surface ligands (MPA), as previously described [50]. TMAH (100 mg) and 22.5 μL of MPA were added to 1 mL of CHCl_3_. The mixture was then incubated at room temperature for 1 h. Subsequently, a clear colorless aqueous layer was formed above the CHCl_3_ layer. The biphasic solution was mixed via vigorous shaking and allowed to equilibrate for 1 h. The organic phase at the bottom was transferred into a vial for the ligand-exchange reaction with CdSe/ZnS QDs. CdSe/ZnS QDs (0.25 mg, dispersed in 100 μL of CHCl_3_) were added to the MPA-CHCl_3_ solution and mixed well. The solution was then allowed to stand at room temperature for 3 h. After the reaction, the MPA-capped QDs separated from the CHCl_3_ solutions were collected, washed with CHCl_3_ (thrice), and dispersed in 1 mL of distilled water (final concentration: 0.025 mg/mL).

### 3.5. Fabrication of SiO_2_@InP QDs@ SiO_2_ NPs with Different Number of Embedded QDs

SiO_2_@InP QDs@SiO_2_ NPs with different numbers of embedded QDs were fabricated by varying the amount of added QDs with two-fold serial dilutions of 5.6, 2.8, 1.4, 0.7, 0.35, 0.175, 0.0875, and 0 mg per 1 mg of thiol-modified SiO_2_ NPs. Thiol-modified SiO_2_ NPs (1 mg in 100 μL EtOH) and 5 μL distilled water were added to 400 μL DCM and mixed with 0 mg to 5.6 mg InP/ZnS QDs (25 mg/mL in toluene). The subsequent fabrication flow for the SiO_2_@InP QDs@SiO_2_ NPs with different numbers of embedded QDs was identical to that described above for SiO_2_@InP QDs@SiO_2_ NPs.

### 3.6. Characterization of SiO_2_@InP QDs@ SiO_2_ NPs

Transmission electron microscopy (TEM) images of SiO_2_@InP QDs@SiO_2_ NPs were obtained using a JEM-2010 system (JEOL, Tokyo, Japan). UV/Vis/NIR absorbance spectra of the SiO_2_@InP QDs@SiO_2_ NPs were obtained using a Optizen Pop UV/Vis spectrophotometer (Mecasys, Daejeon, Korea). Photoluminescence (PL) emission spectra of the SiO_2_@InP QDs@SiO_2_ NPs were obtained using a Cary Eclipse (Agilent Technologies, Santa Clara, CA, USA). The quantum yield (QY) of the SiO_2_@InP QDs@SiO_2_ NPs was measured using a QE-2000 (Otsuka Electronics, Osaka, Japan).

### 3.7. Cytotoxicity Investigation of SiO_2_@InP QDs@ SiO_2_ NPs

A549 cells were maintained in high-glucose DMEM supplemented with 10% FBS, 100 mg/mL streptomycin, and 100 U/mL penicillin, and incubated at 37 °C in 5% CO_2_. A549 cells were seeded in 96-well plates at a density of 8 × 10^3^ cells/well in 100 μL of medium and grown at 37 °C for 18 h. The cells were then treated with two-fold serial diluted SiO_2_@InP QDs@SiO_2_ NPs and hydrophilic CdSe/ZnS QDs (3.56 × 10^11^–1.39 × 10^9^ and 1.96 × 10^14^–7.66 × 10^11^ particles/mL, respectively) in 100 μL medium. After incubation at 37 °C for 24 h, cells were washed with PBS and fixed with 4% paraformaldehyde at room temperature for 2 h. Next, the fixed cells were stained with 0.5% crystal violet and washed thrice with distilled water. The crystal violet-stained cells were de-stained with 1% SDS solution. Absorbance was measured at 585 nm using a VICTOR X3 Multilabel Plate Reader (PerkinElmer, Waltham, MA, USA).

### 3.8. In Vivo Biodistribution of SiO_2_@InP QDs@ SiO_2_ NPs

All animal experiments were approved by the Institutional Animal Care and Use Committee (IACUC) of Konkuk University. To establish tumor syngeneic mice, 1 × 10^6^ 4T1 cells were subcutaneously administered to the BALB/c mice. When the tumor had grown to approximately 400 mm^3^, mice were intravenously administered SiO_2_@InP QDs@SiO_2_ NPs and hydrophilic CdSe/ZnS QDs at a dose equivalent to 200 μL (2.22 × 10^11^ particles/mL) diluted in D5W (dextrose 5% in water) via the tail vein. Mice were euthanized 24 h after intravenous administration, and the major organs (liver, lungs, kidney, spleen, and tumor) were obtained. The biodistributions of SiO_2_@InP QDs@SiO_2_ NPs and hydrophilic CdSe/ZnS QDs were monitored, and the fluorescence intensity of each organ was determined using an IVIS imaging system (Xenogen Corp., Hopkinton, MA, USA).

## 4. Conclusions

In summary, we fabricated highly bright and biocompatible fluorescent nanoprobes, and successfully applied them for bioimaging. SiO_2_@InP QDs@SiO_2_ NPs were fabricated by densely embedding approximately 1200–1500 InP/ZnS QDs onto the surface of the SiO_2_ NPs. The final fabricated SiO_2_@InP QDs@SiO_2_ NPs exhibited a QY of 6.61%. The maximum emission peak wavelength and FWMH of the SiO_2_@InP QDs@SiO_2_ NPs were 527.01 nm, and 44.62 nm, respectively. Even after the SiO_2_@InP QDs@SiO_2_ NPs were fabricated, there were no significant differences between the emission peaks and the FWHM values of the InP/ZnS QDs. In addition, the SiO_2_@InP QDs@SiO_2_ NPs showed good colloidal dispersibility in hydrophilic solvents because of the silica shell on their surface, which is advantageous for biological applications. We also confirmed that the brightness of the SiO_2_@InP QDs@SiO_2_ NPs could be easily controlled by adjusting the amount of added InP/ZnS QDs mixed with thiol-modified SiO_2_ NPs. The fabricated SiO_2_@InP QDs@SiO_2_ NPs selectively accumulated in the tumors of tumor syngeneic mice, with a fluorescence intensity comparable to that of hydrophilic CdSe/ZnS QDs. Thus, SiO_2_@InP QDs@SiO_2_ NPs proved to be useful in biomedical fields with high sensitivity and low toxicity, particularly for bioimaging via in vivo tumor tracking.

## Figures and Tables

**Figure 1 ijms-23-10977-f001:**
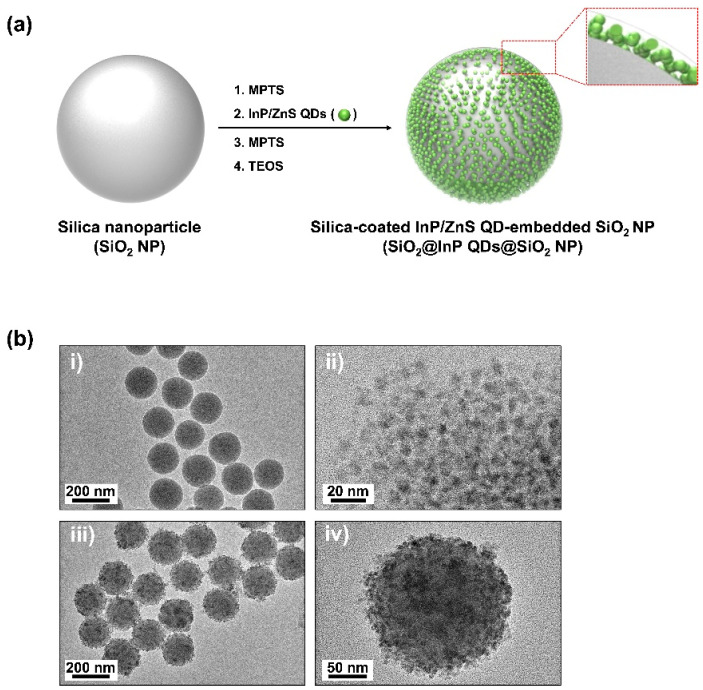
Fabrication of SiO_2_@InP QDs@SiO_2_ NPs. (**a**) Schematic illustration for fabrication of SiO_2_@InP QDs@SiO_2_ NPs. (**b**) Transmission electron microscopy (TEM) images of (**i**) SiO_2_ NPs, (**ii**) InP/ZnS QDs (**iii**), and (**iv**) SiO_2_@InP QDs@SiO_2_ NPs.

**Figure 2 ijms-23-10977-f002:**
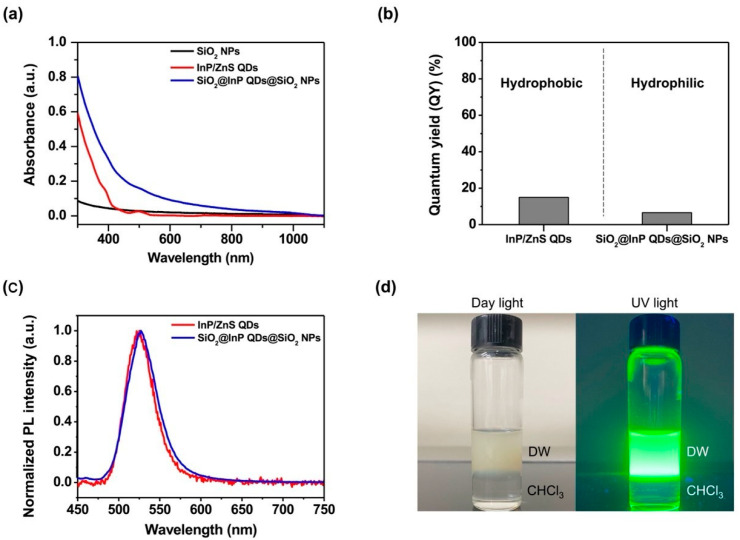
Characterization of SiO_2_@InP QDs@SiO_2_ NPs. (**a**) UV/Vis/NIR absorbance spectra of SiO_2_ NPs, InP/ZnS QDs, and SiO_2_@InP QDs@SiO_2_ NPs. (**b**) Comparison of quantum yield (QY) between InP/ZnS QDs and SiO_2_@InP QDs@SiO_2_ NPs. (**c**) Comparison of photoluminescence (PL) spectra between InP/ZnS QDs and SiO_2_@InP QDs@SiO_2_ NPs. (**d**) Digital images of SiO_2_@InP QDs@SiO_2_ NPs distributed in distilled water (DW).

**Figure 3 ijms-23-10977-f003:**
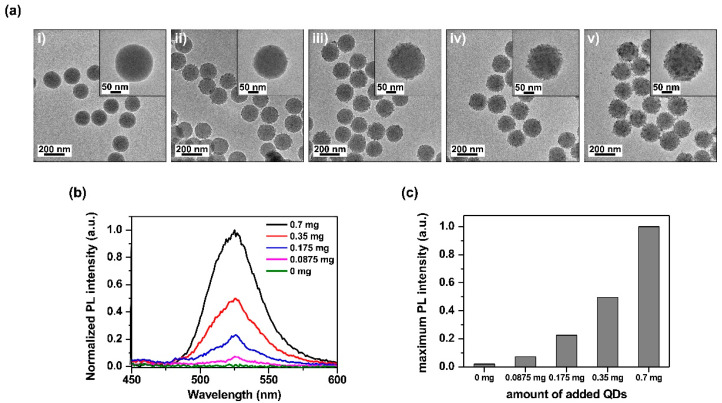
Brightness control of SiO_2_@InP QDs@SiO_2_ NPs. (**a**) TEM image of SiO_2_@InP QDs@SiO_2_ NPs via the amount of added QDs. The amount of added QDs were (**i**) 0 mg, (**ii**) 0.0875 mg, (**iii**) 0.175 mg, (**iv**) 0.35 mg, and (**v**) 0.7mg per 1 mg of SiO_2_ NPs. (**b**) Normalized PL intensity spectra of SiO_2_@InP QDs@SiO_2_ NPs via amount of added QDs. (**c**) Maximum PL intensity of SiO_2_@InP QDs@SiO_2_ NPs via amount of added QDs at 527 nm emission wavelength.

**Figure 4 ijms-23-10977-f004:**
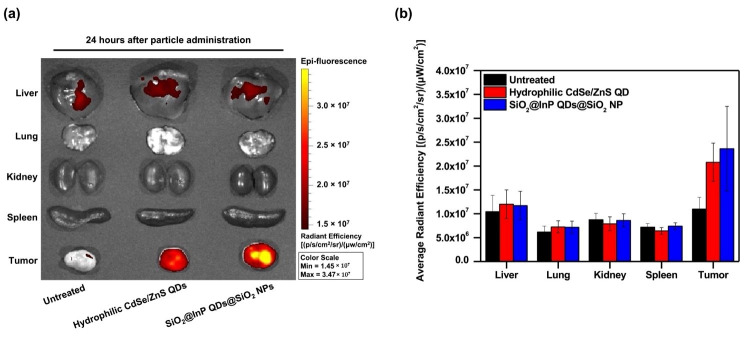
In vivo biodistribution of SiO_2_@InP QDs@SiO_2_ NPs. (**a**) Comparison of biodistribution and fluorescence of particles in major organs and tumors after administration of SiO_2_@InP QDs@SiO_2_ NPs and hydrophilic CdSe/ZnS QDs to tumor syngeneic mice. (**b**) Comparison of average radiant efficiency in major organs and tumors.

## Data Availability

Not applicable.

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
