# Peer review of "Highly Bright Silica-Coated InP/ZnS Quantum Dot-Embedded Silica Nanoparticles as Biocompatible Nanoprobes"

_ijms, 2022, doi:10.3390/ijms231810977_

Round 1
Reviewer 1 Report
In this paper, the authors design a new type of fluorescent nanoparticles based on Cd-free quantum dots incorporated in silica particles. This approach is interesting because allows for the formation of bright and less toxic fluorescent biomarkers compared with CdSe quantum dots. However, some conclusions of the authors are non-clear regarding to the provided experimental details. The main question is why authors compare the photoluminescence intensity of single CdSe QDs with nanospheres containing numerous Cd-free QDs. The more fair comparison may be with silica spheres containing the same amount of CdSe QDs. I believe that a comparison between single CdSe QDs and single silica spheres containing numerous Cd-free QDs can mislead the reader. Thus I encourage authors to remove this comparison and redraw conclusions.
Author Response
Thank you for considering the enclosed manuscript “Highly bright silica-coated InP/ZnS quantum dot-embedded silica nanoparticles as biocompatible nanoprobes” (ijms-1905826) for publication in the International Journal of Molecular Sciences as a research communication.
We appreciate the comments from reviewer who spent invaluable time and effort. We have incorporated additional modifications based on the reviewer’ thoughtful comments, which has helped us to improve the manuscript. The detailed responses to the reviewer’ comments are provided.
Sincerely Yours,
Bong-Hyun Jun
Associate professor
Department of Bioscience and Biotechnology
Konkuk University, 143-701, Seoul, Republic of Korea
Tel: +82-2-450-0521
FAX: +82-2-3437-1977
E-mail: bjun@konkuk.ac.kr

Reviewer 2 Report
My comments and questions refer to the paragraphs in lines 147-172:
1) It sounds not very adequate to compare the emission of hundreds of QD, even though bound to a single silica particle, with that of a single QD. Well, don't you just claim that a large number of QDs with poor PL yield (and lower absorbance) luminesce together brighter that one QD with higher yield?
2) It might be good to provide PL spectrum of the CdSe QDs, not only those of SiO2@InP QDs@SiO2 NPs, to support the statement "They exhibit a photoluminescence (PL) emission wavelength range similar to that of InP/ZnS QDs" (lines 150-151).
3) What is the reason for lower PL quantum yield of QDs embedded in a silica nanoparticle than that of identical free QDs?
4) a minor spelling correction is needed in line 162: "When a light source with an excitation wavelength of 385 nm was irradiated..." - rather not the source was irradiated but QDs were irradiated with the light produced by the source?
Author Response

(The authors gave the same response as above.)

Reviewer 3 Report
In this article the Authors report highly bright silica coated INP/ZnS quantum dots onto silica templets.
The authors reported q quantum dots nanoparticles that exhibited hydrophilic properties because of the surface silica shell, and the maximum PL intensity of the SiO2@InP QDs@SiO2 NPs was approximately 98.4 times higher than that of a single hydrophilic cadmium selenide/ZnS (CdSe/ZnS) QDs. And can be used as highly bright and biocompatible fluorescent nanoprobes for biomedical imaging.
I recommend the manuscript for publishing after minor revisions indicated below.
1. Line 73-74: “However, there has been little progress in the fabrication of silica template-based multi-QDs using InP/ZnS QDs; and their applications in bioimaging”
I will suggest to the authors to add a reference on that, you can add a review or an article, just a few examples that you can find if you look for the keywords.
2. Revise the citations. For example, these articles are related to your topic and are missing in your manuscript.
New J. Chem., 2018,42, 18951-18960, Inorg. Chem. 2021, 60, 9, 6503–6513, RSC Adv., 2018, 8, 25526-25533.
J. Mater. Chem. B, 2018,6, 2574-2583, in this article is relevant to your work even though they are using gold instead of silica for dual-modal imaging with nanoprobe based on gold nanorods and InP/ZnS quantum dots.
3. You are pointing “maximum PL intensity of the SiO2@InP QDs@SiO2 NPs was approximately 98.4 times higher than that of a single hydrophilic cadmium selenide/ZnS (CdSe/ZnS) QDs.” how you calculated this value? I would like to see the comparison because you mentioned it as a novelty in your abstract but you don’t mention it in your conclusions.
Author Response

(The authors gave the same response as above.)

Round 2
Reviewer 1 Report
The authors took into account my comments.
Reviewer 3 Report
Thank you to the authors for incorporating the changes needed in the manuscript. Accepted in present form.